# EFFICIENT ARCHITECTURAL ASPECTS FOR TEXT-TO-VIDEO GENERATION PIPELINE

## ABSTRACT

Multimedia generation approaches occupy a prominent place in artificial intelligence research. Text-to-image models achieved high quality results over the last years, however video synthesis methods recently started to develop. In this paper we present a new two-stage latent diffusion video generation architecture using a new MoVQ video decoding scheme. The first stage concerns keyframes synthesis, while the second one is devoted to interpolated frames generation. We compare two temporal conditioning approaches during evaluation and show the improvement of using temporal blocks over temporal layers in terms of IS and CLIPSIM metrics reflecting video generation quality aspects. We also evaluate different configurations of MoVQ-based video decoding scheme to achieve higher PSNR, SSIM, MSE and LPIPS scores. Finally, we compare our pipeline with existing solutions and achieve top-3 CLIPSIM metric score (0.2976).

## 1 INTRODUCTION

The task of video generation is a natural and logical continuation of the development of generative learning and, in particular, text-to-image generative approaches, which in recent years has achieved stunning results Nichol et al. (2022); Ramesh et al. (2022); Rombach et al. (2022); Saharia et al. (2022). The emergence of diffusion probabilistic models Sohl-Dickstein et al. (2015); Ho et al. (2020); Song et al. (2021) played an important role in the image generation quality improvement. Text-to-video generative diffusion models are also becoming extremely popular, but the problems inherent in this particular task still pose a serious challenge.

Such problems include, among other things, the computational costs of training and inference and the scarcity of large high quality open-source text-video datasets. The available data is not enough to fully understand all the generation possibilities when training from scratch. In addition, such datasets impose restrictions on models related to the specificity of video domains. For this reason, the use of large pretrained text-to-image generative models as the initial step has become the rule of thumb for the video generation field. This allows to transfer comprehensive knowledge of text-to-image models about the visual world to the video world. Also, the use of latent diffusion models Rombach et al. (2022) reduces the volume of computational resources.

However, video generation is not limited to the mentioned difficulties. Video generation is also more complicated than image generation because in order to achieve a high degree of realism and aesthetics, it requires not only the visual quality of a single frame, but also the frame coherence in terms of semantic content and appearance, smooth transitions between frames, as well as correct physics of movements. On the other hand, text-to-image models do not have these requirements. The main key that is responsible here for the mentioned aspects is the temporal information inherent in the video as an object in space-time. Accordingly, the quality of generation will ultimately largely depend on the data processing along the time dimension of video sequences.

As a rule, temporal information is taken into account in diffusion models by including temporal convolutional layers or temporal attention blocks in the architecture Ho et al. (2022b); Wu et al. (2022); Singer et al. (2022); Ho et al. (2022a); Esser et al. (2023); Blattmann et al. (2023); Zhou et al. (2022); Zhang et al. (2023b); Li et al. (2023). This allows initializing the weights of the remaining spatial layers with the values of the weights of the pretrained text-to-image model and training only the temporal layers, which is effective in terms of reducing memory consumption.

In this paper, we separate the text-conditional video generation process based on latent diffusion models into two stages: key frames generation step and frames interpolation step. The key frames are designed to set the main amount of semantic information for the future video. This separation allows us to maintain accordance with the text description along the entire length of the video in terms of both content and dynamics. At the stage of key frames generation, we compare temporal conditioning approaches, namely using traditional mixed spatial-temporal blocks and the proposed separate temporal blocks. We find that the use of the latter makes it possible to significantly improve the video quality both in terms of visual aesthetics and dynamics. We propose this solution as a general approach to include temporal components in text-to-image models to use them in video generation. For the interpolation step, we also modify the text-to-image model and claim that the use of context guidance is a key factor for smooth transitions between frames. At the experimental stage we are also exploring various possibilities of including temporal architecture components in the MoVQGAN Zheng et al. (2022) decoder to improve the quality of latent video generation.

Thus, our contribution contains the following aspects:

- We present an end-to-end text-to-video latent diffusion pipeline, which is based on the pretrained frozen text-to-image model. Our pipeline divided into two parts – key frames generation and frames interpolation.

- As a part of the key frames generation, we propose to use separate temporal blocks to account for temporal information. We compare this approach with incorporated temporal layers and demonstrate the qualitative and quantitative advantage of our solution in terms of visual quality and temporal consistency using a set of metrics (IS, CLIPSIM) on several video datasets in different domains.

- We provide an overview of our interpolation architecture, which incorporates temporal output masking in conjunction with data augmentations, contributing to robust video frame interpolation.

- We investigate various architectural options to build Video Decoder and evaluate their performance in terms of quality metrics and the impact on the size of decoder.

## 2 RELATED WORK

### 2.1 TEXT-TO-VIDEO GENERATION

Prior works on video generation utilize VAEs Mittal et al. (2017); Babaeizadeh et al. (2017; 2021); Yan et al. (2021); Walker et al. (2021), GANs Vondrick et al. (2016); Pan et al. (2017); Li et al. (2018); Lee et al. (2018); Clark et al. (2019), normalizing flows Kumar et al. (2019) and autoregressive transformers Wu et al. (2021a;b); Ge et al. (2022); Hong et al. (2022); Villegas et al. (2022). GODIVA Wu et al. (2021a) adopts a 2D VQVAE along with sparse attention for Text-to-Video generation. CogVideo Hong et al. (2022) is built on top of a frozen CogView2 Ding et al. (2022) text-to-image transformer by adding additional temporal attention layers.

Recent research extend text-to-image diffusion-based architecture for text-to-video generation Singer et al. (2022); Ho et al. (2022a); Blattmann et al. (2023); He et al. (2022); Zhou et al. (2022). This approach can benefit from pretrained image diffusion models and transfer that knowledge to video generation tasks. Specifically, it involves the introduction of temporal convolution and attention layers interleaved with existing spatial layers. This adaptation aims to capture temporal dependencies between video frames while also achieving computational efficiency by avoiding the use of infeasible 3D convolutions and 3D attention mechanisms. These temporal layers can be trained independently or jointly with the 2D spatial layers.

The first widely known end-to-end model in which this technique was applied was VDM (Ho et al. (2022b)) based on the 3D UNet, in which $3 \times 3$ convolutions were replaced with $1 \times 3 \times 3$ convolutions, and after each spatial attention layer, a temporal attention layer was added. In Make-a-Video approach (Singer et al. (2022)) each spatial 2D convolution was followed by a temporal 1D convolution. This technique of mixed spatial-temporal blocks for both convolutions and attention layers has become widespread in most subsequent text-to-video models (Ho et al. (2022a); Esser et al. (2023); Wu et al. (2022); Blattmann et al. (2023); Zhou et al. (2022); Li et al. (2023)). Alternative

approaches to operate with the time dimension include the use of image diffusion model in conjunction with a temporal autoregressive Recurrent Neural Network (RNN) model to predict individual video frames (Yang et al. (2022)), projection of 3D data into a latent 2D space (Yu et al. (2023)) and the use of diffusion to generate latent flow sequences (Ni et al. (2023)). We propose a new approach for conditioning on temporal information and instead of blocks with spatial and temporal layers, we consider separate temporal blocks.

Finally, it is worth noting that certain studies (Singer et al. (2022); Ho et al. (2022a)) operate entirely in pixel space, whereas others He et al. (2022); Zhou et al. (2022); Wu et al. (2022); Esser et al. (2023); Blattmann et al. (2023); Zhang et al. (2023b); Li et al. (2023)) utilize the more efficient latent space. We follow the second approach in this paper.

## 2.2 VIDEO FRAME INTERPOLATION

In text-to-video generation, the typical approach involves a two-stage process: first generating keyframes and then interpolating between these generated keyframes. This method simplifies the overall generation, ensuring coherence and smoothness, minimizing artifacts, and improving realism.

Existing video frame interpolation techniques (Zhang et al. (2023a); Reda et al. (2022)) are primarily designed for real video decompression and often prove inadequate when applied to interpolating between synthesized keyframes. Our examination of these models revealed specific failure scenarios, notably when dealing with large object motion and occlusion. Utilizing these methods for interpolation frequently results in undesirable blurring and ghosting artifacts in the generated frames which affect the realism of the final video.

MCVD (Voleti et al. (2022)) utilizes a diffusion-based model for interpolation, leveraging the two keyframes from each side to generate three frames in between. Within Text-to-Video research, a diffusion-based T2V architecture is employed for frame interpolation (Blattmann et al. (2023); Ho et al. (2022a); Zhou et al. (2022)). Two keyframes are combined with a noisy input in a channelwise concatenation before being fed into the UNet architecture. The entire UNet-based model is subsequently trained to predict one or more frames between each pair of keyframes.

Consistent with prior researches, we incorporate conditioning frames from both the left and right keyframes to generate three middle frames. We also adopt temporal layers to enhance temporal coherence. Furthermore, we include the conditioning frame perturbation technique, combined with augmentations, to enhance the model's resilience in situations where the generated keyframes are less than optimal. This leads us to present our ultimate architecture for robust video frame interpolation.

## 2.3 VIDEO DECODER

Utilizing an Image decoder for frame decoding often leads to inconsistencies in the resulting details and the presence of flickering artifacts. To address this issue and ensure a more uniform generation process, the integration of temporal layers becomes essential. In their works, Blattmann et al. (2023) builds a video decoder with 3D convolution, while Li et al. (2023) enhances it with temporal 1D convolution and temporal self-attention. Zhou et al. (2022), on the other hand, incorporates two temporal directed attention layers in the decoder to build a VideoVAE decoder. To the best of our knowledge, previous studies have not offered a comparison of their strategies for constructing a video decoder. In this research, we present multiple options for designing a video decoder and conduct an extensive comparative analysis, evaluating their performance in relation to quality metrics and the implications for additional parameters.

## 3 DIFFUSION PROBABILISTIC MODELS

Denoising Diffusion Probabilistic Models (DDPM) (Ho et al. (2020)) is a family of generative models designed to learn a target data distribution $p_{data}(x)$. It consists of a forward diffusion process and a backward denoising process. In the forward process, random noise is gradually added into the data $x$ through a $T$-step Markov chain (cite). The noisy latent variable at step $t$ can be expressed as:

$$z_t = \sqrt{\hat{\alpha}_t}x + \sqrt{1 - \hat{\alpha}_t}\epsilon \qquad (1)$$

with $\hat{\alpha}_t = \prod_{k=1}^{t} \alpha_k, 0 \leq \alpha_k < 1, \epsilon \sim N(0,1)$. For a sufficiently large $T$, e.g., $T = 1000$, $\sqrt{\hat{\alpha}_T} \approx 0$, and $1 - \sqrt{\hat{\alpha}_T} \approx 1$. Consequently, $z_T$ ends up with pure noise. The generation of $x$ can then be seen as an iterative denoising process. This denoising process corresponds to learning the inverted process of a fixed Markov Chain of length T.

$$L_t(x) = \mathbb{E}_{\epsilon \sim N(0,1)}[\|\epsilon - z_\theta(z_t, t)\|_2^2] \qquad (2)$$

Here, $z_\theta$ represents a denoising neural network parameterized by $\theta$, and $L_t$ is the loss function.

## 4 METHODS

**Overall pipeline.** The scheme of our text-to-video model is shown in the Fig 1. It includes a text encoder, a keyframe latent generation model, a frame interpolation model, and a latent decoder that is the single model, but performs differently when decoding keyframes and full video. Below we describe the key components in detail.

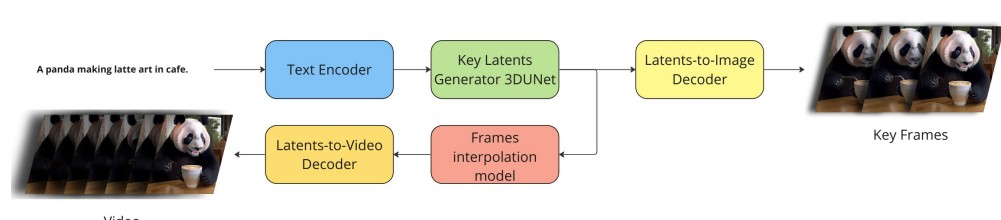

Figure 1: **The overall scheme of the pipeline.** The encoded text prompt gets into the UNet keyframe generation model with temporal layers or blocks, and then the sampled keyframes are putting into the latent interpolation model in such a way as to predict three interpolation frames between two keyframes. A temporal MoVQ-GAN decoder is used to get the final video result.

### 4.1 KEYFRAMES GENERATION WITH TEMPORAL CONDITIONING

The keyframes generation is based on a pre-trained latent diffusion text-to-image model. We use the weights of this model to initialize the spatial layers of the keyframe generation model, which is distinguished by the presence of temporal components. In all experiments, the weights of text-to-image UNet remained frozen, and only temporal components were trained.

We are considering two key ways of introducing temporal components into architecture – using temporal layers of convolutions and attention, as well as using our separate temporal blocks. The Fig. 2 exhaustively explains our concept.

The traditional approach with a mixed spatial-temporal block is the inclusion of temporal components of architecture in the environment of spatial layers. So, in the case of convolutions (Fig. 2 (a), left), after a spatial convolution with a $3 \times 3$ kernel (dimensions correspond to the high and width of the frame), a $3 \times 1 \times 1$ temporal 1D convolution follows, in which the first dimension of the kernel corresponds to the time axis along video frames in the batch. In Fig. 2 (a) on the right is our separate temporal block, in which all temporal convolutions are located, and the gray frozen spatial block remained unchanged in comparison with the block of UNet from the text-to-image model. The Fig. 2 (b) shows a similar case for the spatial and temporal attention.

### 4.2 VIDEO FRAME INTERPOLATION

In the latent space, interpolation is applied to predict three middle frames between each pair of keyframes. This necessitates the adaptation of the text-to-image architecture. Changes are made to

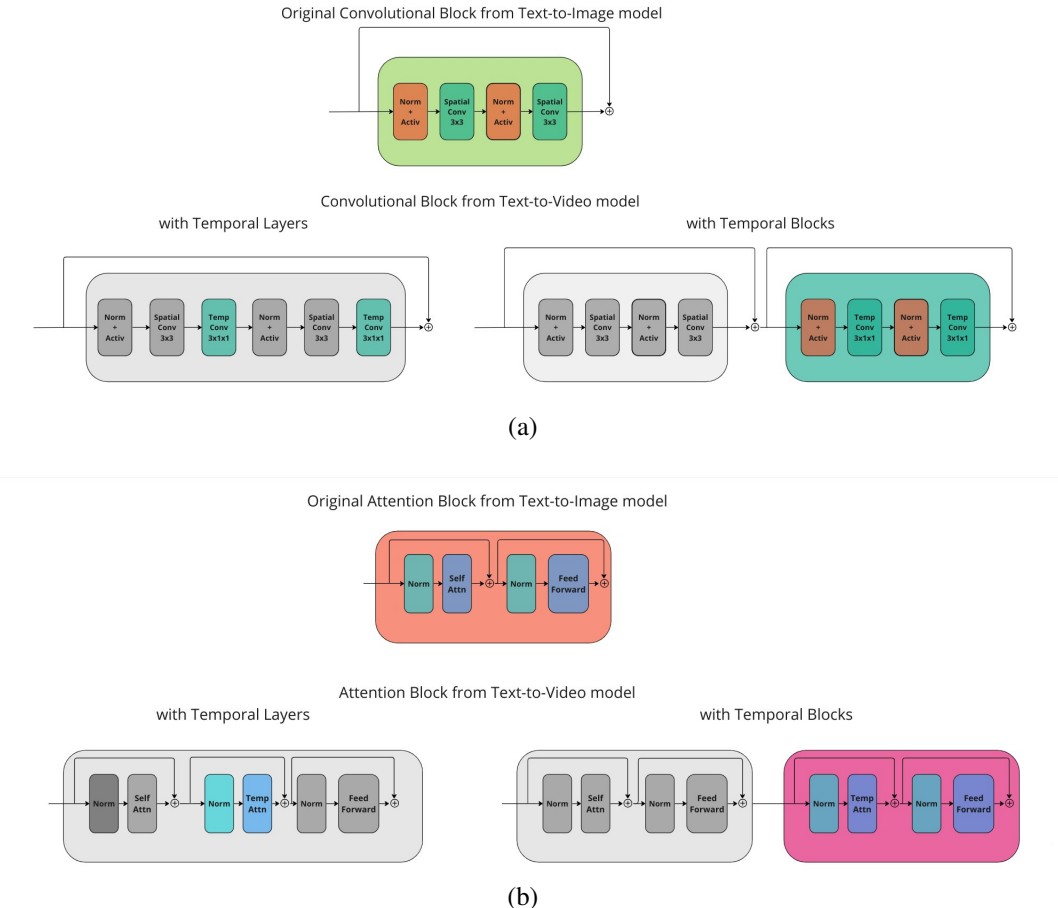

Figure 2: **Comparative schemes for temporal conditioning.** We compare two approaches for including temporal components of architecture in a pre-trained text-to-image UNet – the traditional approach of mixing spatial and temporal layers in one block **(left)** and our approach of allocating a separate temporal block **(right)**. All layers indicated in gray are not trained in text-to-video architectures and are initialized with the values of the weights of the text-to-image model. **a)** A case for convolutions, **b)** a case for attention.

the input convolution layer, enabling it to handle three noisy input frames $z$ with two conditioning frames $c$ concatenated all together. The output convolution layer is similarly adjusted to generate three middle frames. Temporal convolution layers $\phi$ are also introduced after each spatial convolution in the original text-to-image model $\theta$. A detailed description of these adjustments is provided in Appendix A. As part of this process, text conditioning is removed, and instead, we incorporate skip-frame $s$ and perturbation noise level conditioning $pt$. For interpolation, we use v-prediction parameterization ($v_t \equiv \alpha_t \epsilon - \sigma_t x$) as described in (Salimans & Ho (2022); Ho et al. (2022a)).

Building on the conditioning frame perturbation technique (Blattmann et al. (2023); He et al. (2022)), we randomly sample a perturbation level $tp \in \{0, 1, 2, \ldots, 250\}$. This $tp$ value guides in perturbing the conditioning keyframes using a variance-preserving diffusion process, following the cosine diffusion schedule utilized in the main diffusion model. This perturbation noise level is also employed as a condition for the UNet. Additionally, with a some probability $uncond\_prob$, we replace the conditioning frames with zeros and zero out the temporal layer outputs using a mask $mt$ for unconditional frames generation training. $mt = 0$ implies that we mask the output produced by the temporal convolution layers. The final training objective looks like:

$$L_t(x; tp, s, mt) = \mathbb{E}_{\epsilon \sim N(0,1)}[\|v - z_{\theta,\phi}(z_t, t, c, tp, s, mt)\|_2^2] \qquad (3)$$

We employ context guidance (Blattmann et al. (2023)) at inference, with $w$ representing the guidance weight:

$$\tilde{z}_{\theta,\phi}(z_t, c, tp, s) = (1 + w)z_{\theta,\phi}(z_t, c, tp, s, mt = 1) - wz_{\theta,\phi}(z_t, s, mt = 0) \quad (4)$$

Our interpolation model can benefit from a relatively low guidance weight value, such as 0.25 or 0.5. Increasing this value can have a significant negative impact on the quality of the interpolated frames, sometimes resulting in the model generating frames that closely match the conditioning frames.

### 4.3 VIDEO DECODER

Modulated VQGAN (Zheng et al. (2022)) is an advanced variant of the Vector Quantized Generative Adversarial Network (VQGAN) Esser et al. (2020). It enhances image generation by using spatially conditional normalization to reduce artifacts in adjacent regions and employs multichannel quantization for improved code recombination resulting in high-fidelity, photo-realistic images.

To enhance the video decoding process, we make use of a pretrained MoVQ-GAN model[1] with a frozen encoder. To extend the decoder into the temporal dimension, we explore several choices: The substitution of 2D convolutions with 3D convolutions and the addition of temporal layers interleaved with existing spatial layers. Regarding the temporal layers, we explore the use of temporal convolutions, temporal blocks and temporal self-attentions. All additional parameters are initialized with zeros.

This extension is essential in improving consistencies of generated details among frames, reducing flicker artifacts, and as result improving the fidelity of the resulted video.

## 5 EXPERIMENTS

### 5.1 EXPERIMENTAL SETUP

**Datasets.** Our internal training dataset for Text-to-Video contains $120k$ text-video pairs, and the same dataset is utilized for training the interpolation model. In our evaluation, we evaluate the T2V model on two testsets: UCF-101 (Soomro et al. (2012)) and MSR-VTT (Xu et al. (2016)). For training the decoder, we use a mix of $80k$ videos from the internal dataset, while model testing is performed on the subtuplet part of Vimeo90k (Xue et al. (2019)) dataset.

In preparing frame sequences for the interpolation task training, we randomly select a skip-frame value $s \in \{1, 2, \ldots, 12\}$. Subsequently, the input video is resampled with a skip of $s$ frames between each pair. This resampled video is then organized into 8 conditioning frames for each side and 8x3 target frames, resulting in a total of 33 frames per input video. Augmentation techniques are applied to the condition frames, including the averaging of two consecutive frames and the introduction of blur. Furthermore, we apply consistent augmentations to the entire sequence of frames, such as random flipping, reversing, random cropping, and sliding. For decoder training, sequences comprising 8 frames are employed.

**Metrics.** In line with previous works (Singer et al. (2022); Luo et al. (2023); Li et al. (2023)), we assess our text-to-video model using the following evaluation metrics: Inception Score (IS) (Saito et al. (2020)) and CLIPSIM (Wu et al. (2021a)). IS metric assesses the quality and diversity of individual frames. CLIPSIM, on the other hand, evaluates text-video alignment. For decoder training, other metrics are used: PSNR for frame quality assessment, SSIM for structural similarity evaluation and LPIPIS[2] (Zhang et al. (2018)) for perceptual similarity.

**Training.** We trained the keyframe generation model for $100k$ steps on 16 GPUs (A100 80GB) with batch size of 1, gradient accumulation of 2 to generate 8 frames in $512 \times 512$ resolution. We used dynamic FPS by encoding the positions of generated frames scaled by this FPS. We train only temporal layers or temporal blocks depending on the method. All other weights have been taken from our text-to-image model.

---

[1] https://github.com/ai-forever/MoVQGAN
[2] https://github.com/richzhang/PerceptualSimilarity

The entire interpolation model is trained for $50k$ steps, at the task of upsampling 9 frames across different skip-frame values $s$ to a sequence of 33 frames, all while maintiaining a resolution of 256x256. During training, we set the probability for unconditional frame generation $uncond\_prob$ to $10\%$. Our decoder, including the spatial layers, is trained using sequences of 8 frames for $50k$ steps. During training, we use 8 GPUs (A100 80GB) and gradient accumulation steps of $4$. For decoder training, we turn off gradient accumulation.

**Inference.** Keyframes are generated in the text-to-video generation phase. To interpolate between these generated keyframes, we employ the generated latents from the first phase as conditions to generate three middle frames between each keyframe pair. We set the skip-frame value $s$ to 6 during this stage of interpolation. Additionally, we maintain a constant perturbation noise level $tp$, with a value of 150, without actually perturbing the conditioning frames. In cases where unconditional generation is applied as part of the context guidance approach, we adjust $pt$ value to zero and we set the guidance weight to a small value $w = 0.25$. In the final stage, our trained decoder decodes the resulting latents together with the latents from the generated keyframes to produce the final video output.

Visually, the results can be assessed in the Figures 3, 4

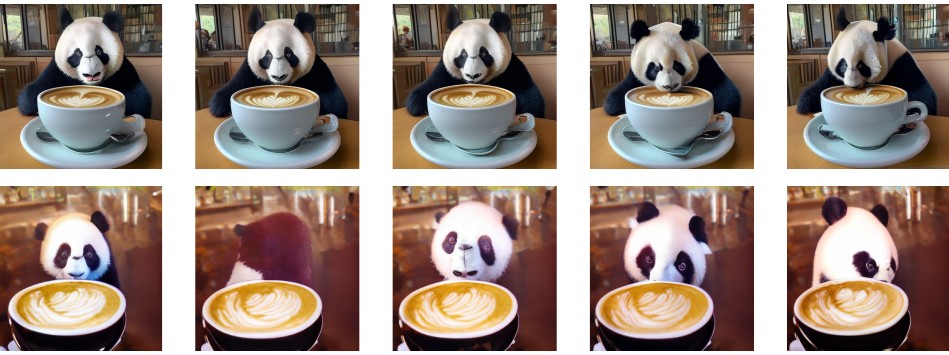

Figure 3: **Prompt**: A panda making latte art in cafe. Bloks up, layers down.

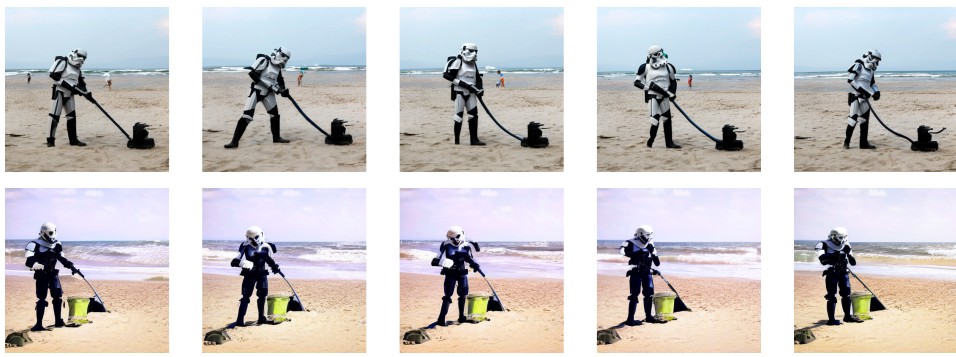

Figure 4: **Prompt**: A storm trooper vacuuming the beach. Bloks up, layers down.

## 5.2 QUANTITATIVE RESULTS

In this section, we provide a comparison of our trained models Using Inception score (IS) on UCF-101 and Clip Similarity (CLIPSIM) on MSR-VTT, as detailed in Table 1 and Table 2 respectively. Our results indicate that the inclusion of temporal blocks, rather than temporal layers, leads to improved quality in terms of these metrics. The IS is lower than baselines due to insufficient training time.

As for the FVD metric Unterthiner et al. (2019), we could not clearly interpret its results, since we did not find a correlation between the actual quality of the generated videos and the numbers that

Table 1: T2V zero-shot results on UCF-101.

| Method | Resolution | IS↑ |
|---|---|---|
| CogVideo (Chinese) | 480x480 | 23.55 |
| CogVideo (English) | 480x480 | 25.27 |
| Make-A-Video | 256x256 | 33.00 |
| VideoGen | 256x256 | **77.61** |
| Ours (temporal layers) | 512x512 | 23.50 |
| Ours (temporal blocks) | 512x512 | **24.17** |

Table 2: T2V results on MST-VTT. We report average CLIPSIM scores.

| Method | Zero-Shot | Resolution | CLIPSIM↑ |
|---|---|---|---|
| GoDIVA | No | 128x128 | 0.2402 |
| Nuwa | No | 336x336 | 0.2439 |
| CogVideo (Chinese) | Yes | 480x480 | 0.2614 |
| CogVideo (English) | Yes | 480x480 | 0.2631 |
| Make-A-Video | Yes | 256x256 | 0.3049 |
| VideoGen | Yes | 256x256 | **0.3127** |
| Ours (temporal layers) | Yes | 512x512 | 0.2904 |
| Ours (temporal blocks) | Yes | 512x512 | **0.2976** |

the estimator gave out for both temporal layers and blocks methods. We have decided to abandon the use of this metric for evaluation.

## 5.3 QUALITATIVE RESULTS

A qualitative comparison of the method of inserted temporal layers along spatial layers with the separate temporal blocks approach reveals visually observable advantages of our technique both in terms of the quality of generated objects on separately taken keyframes, and from the point of view of dynamics Fig. 3, 4. The method based on temporal layers either produces semantically distant keyframes, or does not cope well with the dynamics in some cases. On the contrary, the method based on temporal blocks generates more consistent in content and coherent in time keyframes. There is a general improvement in the quality of generation and a decrease in the number of artifacts compared to mixed spatial-temporal blocks with temporal layers inserted in them.

## 5.4 VIDEO MOVQ-GAN ABLATION STUDY

We conducted comprehensive experiments, considering many choices of how to build video decoder, and assessed them in terms of quality metrics and the impact on the number of additional parameters. The results are presented in table 3. This evaluation process guided us in making the optimal selection for production purposes. Extending the decoder with a 3x3x3 temporal convolution and incorporating temporal attention during the fine-tuning process, which applies to the entire decoder, including spatial layers and the newly introduced parameters, yields the highest overall quality among the available options. An alternative efficient choice involves using a 3x1x1 temporal convolutional layer or temporal ResNet Block with temporal attention, which significantly reduces the number of parameters from $556M$ to $220M$ while still achieving results that closely match the quality obtained through the more extensive approach.

Table 3: A comparison between different choices to construct video decoder including the use of temporal convolution, temporal ResNet Block, temporal attention (Attn) and finally converting 2D spatial convolution in the decoder into 3D conv (2D→3D Conv). We also present whether we only finetune temporal layers or the entire decoder.

| Decoder | Temporal Layers | Finetune | PSNR↑ | SSIM↑ | MSE↓ | LPIPS↓ | # Params |
|---------|-----------------|----------|-------|-------|------|--------|----------|
| Image | - | - | 32.9677 | 0.9056 | 0.0008 | 0.0049 | 161 M |
| Video | 3x1x1 Conv | Temporal | 32.2544 | 0.893 | 0.0009 | 0.006 | 203 M |
| Video | 3x3x3 Conv | Temporal | 33.5819 | 0.9111 | 0.0007 | 0.0044 | 539 M |
| Video | 3x1x1 Conv | Decoder | 33.5051 | 0.9106 | 0.0007 | 0.0044 | 203 M |
| Video | 3x3x3 Conv | Decoder | 33.6342 | 0.9123 | 0.0007 | 0.0043 | 539 M |
| Video | 3x1x1 Conv + Attn | Decoder | 33.7343 | 0.9129 | 0.0007 | 0.0043 | 220 M |
| Video | 3x3x3 Conv + Attn | Decoder | **33.8376** | **0.9146** | **0.0006** | **0.0041** | 556 M |
| Video | ResNet Block + Attn | Decoder | 33.7024 | 0.9121 | 0.0007 | 0.0043 | 220 M |
| Video | 2D → 3D Conv | Decoder | 33.7321 | 0.9134 | 0.0007 | 0.0043 | 419 M |

## 6 CONCLUSION

In this research we examined several ways of the text-to-video generative architecture design in order to get better output quality improvement. This challenging task included the development of a two-stage model for video synthesis taking into account several ways to include temporal information: temporal blocks and temporal layers. According to experiments, the first approach lead to higher metrics values in terms of visual quality measured by IS score. We achieved a comparable IS score value to several existing solutions and a top-3 score in terms of CLIPSIM metric. A new MoVQ-based video decoding scheme is presented in the paper as well as the results of its experimental study. The following problems still remain: image quality of frames should be improved, subsequent frames consistency in terms of smoothness and visual quality preserving due to temporal correlation should be improved using new approaches to latent noise generation and latent features interpretation, that we plan to research further. We also plan to evaluate the quality of open-source text-to-video models with human evaluation and provide some interpretation of Frechet Video Distance(FVD) score, which shows contradictory values on existing research results.

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

## A  INTERPOLATION

Expanding the text-to-image architecture to effectively handle and generate a sequence of interpolated frames requires a series of modifications designed to process the data across the temporal dimension. In addition to this, further adjustments are necessary to facilitate the generation of three middle frames between two keyframes. Specifically, starting with pre-trained weights for the text-to-image model, we replicate the weights within the output convolution layer three times. This transformation alters the dimensions from $(out\_channels, input\_channels, 3, 3)$ to $(3 * out\_channels, input\_channels, 3, 3)$, and a similar modification is carried out for the bias parameters, shifting them from $(out\_channels)$ to $(3 * out\_channels)$. This adaptation enables the generation of three identical frames before the training phase starts. In the input convolution layer, we make similar adjustments to the input channels, initializing additional weights as zeros. Subsequently, a temporal convolution layer, with a kernel size of $(3, 1, 1)$, is introduced after each spatial convolution layer. The output from each temporal layer is then combined with the output from the spatial layer using a learned merge parameter $\alpha$. The spatial layers are designed to handle input as a batch of individual frames. When dealing with video input, a necessary adjustment involves reshaping the data to shift the temporal axis into the batch dimension. Consequently, before forwarding the activations through the temporal layers, a transformation is performed to revert the activations back to its original video dimensions.

