# OpenReview forum: "Efficient architectural aspects for text-to-video generation pipeline"
_ICLR.cc/2024/Conference — ICLR 2024 Conference Withdrawn Submission_

### Official Review · Reviewer_ubtc · 2023-10-15

**Soundness:** 2 fair
**Presentation:** 2 fair
**Contribution:** 2 fair
**Rating:** 5
**Confidence:** 5

**Summary:**

This paper proposes a video generation pipeline based on MoVQ video decoding scheme. It consists of two stages: keyframes synthesis and video frame interpolation. It also compares two temporal conditioning approaches and different configurations of MoVQ-based models.

**Strengths:**

The contributions of this method are summarised as below:

1. an end-to-end text-to-video latent diffusion pipeline that consists of key frames generation and frame interpolation

2. separate temporal blocks for temporal modelling

3. temporal output masking and data augmentations for robust VFI

4. investigation of video decoders

**Weaknesses:**

My concern is the lack of novelty. The concert comments are below:

1. This pipeline is not new. Various methods have tried to utilize text2image diffusion models for video generation. For example, the proposed temporal conditional scheme could be found at ``AnimateDiff: Animate Your Personalized Text-to-Image Diffusion Models without Specific Tuning
''.

2. Concatenating key frames along the channel dimension in video frame interpolation part is similar to "IMAGEN VIDEO: HIGH DEFINITION VIDEO
GENERATION WITH DIFFUSION MODELS".

3. The introduction of temporal layers is similar to "Align your Latents: High-Resolution Video Synthesis with Latent Diffusion Models".

4. What does "efficient" in the title mean?

**Questions:**

See weakness.

---

### Official Review · Reviewer_R6HA · 2023-10-25

**Soundness:** 3 good
**Presentation:** 3 good
**Contribution:** 3 good
**Rating:** 5
**Confidence:** 3

**Summary:**

This paper proposes a new text-to-video generation method for temporal generation. New architecture using keyframe generation and frame interpolation with video decoder is proposed to handle the text-to-video scenario. Some sota results are acheived in terms of IS and CLIPSIM metrics.

**Strengths:**

A novel scheme for video generation with customizations to ordinary text-to-video models.
Evaluated several configurations for temporal blocks and MoVQ-based decoder.

**Weaknesses:**

Not the best results are acheived in terms of CLIPSIM metrics.
IS scores are not good compared with videogen.

**Questions:**

Why the IS score is much lower than videogen ?

---

### Official Review · Reviewer_qUnt · 2023-10-31

**Soundness:** 2 fair
**Presentation:** 3 good
**Contribution:** 2 fair
**Rating:** 5
**Confidence:** 4

**Summary:**

The paper proposes a two-stage latent diffusion model, which contains keyframe generation and frame interpolation, for text-to-video generation tasks. Starting from a pretrained text-to-image model, separate temporal blocks are used to model the temporal information. A video decoder based on MoVQGAN is also adopted to improve the generation quality.

**Strengths:**

1. The topic of this paper is significant.
2. The paper is overall clear and well-formulated.
3. The training of the proposed model seems resource-efficient compared to most text-to-video methods, as only 8-16 A100 GPUs and 120k data pairs were adopted in training.

**Weaknesses:**

1. There needs to be more explanations and analysis about the proposed methods and results. The authors should illustrate the possible reason why separate temporal blocks perform better than temporal layers.
2. As the limitations of quantitative metrics, more visual results should be provided and compared with other text-to-video methods.
3. The paper doesn't mention the number of parameters of the video generation model, and only 120k internal data pairs are utilized for training. The model maybe overfit to the training data. As the data domain is also agnostic, it's hard to decide whether the experiment results are evidential enough or not.

**Questions:**

1. Could the authors give more explanations about the separate temporal blocks over temporal layers?
2. As mentioned in the introduction part, the scarcity of open-source text-video datasets impedes the development of video generation. Is it possible to contribute the internal training data to the community?